# Surveillance of Influenza and Other Airborne Transmission Viruses during the 2021/2022 Season in Hospitalized Subjects in Tuscany, Italy

**DOI:** 10.3390/vaccines11040776

**Published:** 2023-03-31

**Authors:** Giovanna Milano, Elena Capitani, Andrea Camarri, Giovanni Bova, Pier Leopoldo Capecchi, Giacomo Lazzeri, Dario Lipari, Emanuele Montomoli, Ilaria Manini

**Affiliations:** 1Department of Life Science, University of Siena, 53100 Siena, Italy; 2Department of Molecular and Developmental Medicine, University of Siena, 53100 Siena, Italyilaria.manini@unisi.it (I.M.); 3Emergency and Transplants Department, University Hospital of Siena, 53100 Siena, Italy; 4Department of Medical Sciences, Surgery and Neurosciences, University of Siena, 53100 Siena, Italy; 5Interuniversity Research Centre on Influenza and Other Transmissible Infections (CIRI-IT), 16132 Genoa, Italy; 6VisMederi S.r.l., 53100 Siena, Italy

**Keywords:** influenza, COVID-19, co-infections, severe acute respiratory infection, airborne pathogens

## Abstract

Winter in the northern hemisphere is characterized by the circulation of influenza viruses, which cause seasonal epidemics, generally from October to April. Each influenza season has its own pattern, which differs from one year to the next in terms of the first influenza case notification, the period of highest incidence, and the predominant influenza virus subtypes. After the total absence of influenza viruses in the 2020/2021 season, cases of influenza were again recorded in the 2021/2022 season, although they remained below the seasonal average. Moreover, the co-circulation of the influenza virus and the SARS-CoV-2 pandemic virus was also reported. In the context of the DRIVE study, oropharyngeal swabs were collected from 129 Tuscan adults hospitalized for severe acute respiratory infection (SARI) and analyzed by means of real-time polymerase chain reaction (RT-PCR) for SARS-CoV-2 and 21 different airborne pathogens, including influenza viruses. In total, 55 subjects tested positive for COVID-19, 9 tested positive for influenza, and 3 tested positive for both SARS-CoV-2 and the A/H3N2 influenza virus. The co-circulation of different viruses in the population requires strengthened surveillance that is no longer restricted to the winter months. Indeed, constant, year-long monitoring of the trends of these viruses is needed, especially in at-risk groups and elderly people.

## 1. Introduction

Every year, there are two seasonal influenza epidemics: one in the southern and one in the northern hemisphere, since low temperatures and humidity favor the circulation of influenza viruses [1].

Influenza is one of the world’s most contagious seasonal respiratory diseases and is caused by negative-sense single-strand RNA viruses with a segmented genome [1,2]. There are four types of influenza viruses: A, B, C, and D. Viruses A and B are mainly responsible for seasonal influenza. The A viruses are divided into subtypes on the basis of their surface glycoproteins HA (hemagglutinin) and NA (neuraminidase); 18 subtypes of HA and 11 subtypes of NA are now known, and the different combinations of these glycoproteins generate an elevated number of influenza A subtypes [3]. Furthermore, since animals are the reservoir of influenza A viruses, and surface antigens mutate rapidly, type A viruses are more prone to causing pandemics [4,5]. Currently, the A/H1N1 and A/H3N2 subtypes are circulating most widely among humans. 

B viruses are divided into lineages. Currently, the B/Yamagata and B/Victoria lineages are circulating in humans. The time from infection to illness, known as the incubation period, is usually about two days but can range from one to four days [6]. Symptoms can be mild to very severe, depending on the characteristics of the subject, and can sometimes develop into SARI (severe acute respiratory infection). The term SARI is used with reference to a hospitalized patient of any age with at least: one respiratory sign or symptom (cough, sore throat, breathing difficulties) present at the time of hospital admission or within 48 h after admission, and one systemic sign or symptom (fever or low-grade fever, headache, myalgia, generalized malaise) or deterioration in their general condition (asthenia, weight loss, anorexia or confusion, and dizziness) [7]. However, many viruses that cause pulmonary symptoms, including SARS-CoV-2, spread by means of airborne transmission. 

The SARS-CoV-2 virus belongs to the family of Coronaviridae [8] and circulated worldwide before March 2020. On 11 March 2020, the disease caused by SARS-CoV-2, known as COVID-19, was declared a pandemic by the WHO (World Health Organization) [9]. At the time of this writing, the WHO estimates that COVID-19 has been diagnosed in 756,581,850 people, resulting in 6,844,267 deaths. In Italy, as of today [10], 25,519,067 confirmed cases of COVID-19 have been recorded, as has an average of 8000 deaths per year due to influenza and its complications [11]. Human respiratory syncytial virus (HRSV), human parainfluenza virus (hPIV), and human metapneumovirus (hMPV) are associated with chronic lung dysfunction. hPIV is a common cause of hospitalization in children of pre-school age, placing a heavy burden on this age group [12]. The human coronaviruses HCoV-229E, HCoV-OC43, HCoV-NL63, and HKU1 generally cause upper respiratory tract infections and mainly affect at-risk populations. They cause few serious infections, unlike SARS-CoV-2, which also affects the lower respiratory tract and can lead to serious respiratory complications [13]. While human parechoviruses (HPeVs) mostly cause asymptomatic infections, they can result in meningitis and sepsis, especially in infants. By contrast, *Mycoplasma pneumoniae* is an important bacterial pathogen and is responsible for respiratory tract infections in every age group. *Mycoplasma pneumoniae* infection mimics SARS-CoV-2 infection in that it causes pharyngitis, coughing, and dyspnea. Co-infection of hMPV and SARS-CoV-2 can exacerbate clinical symptoms and increase mortality [14]. As the clinical onset of the above-listed infections is similar, with bronchiolitis, upper respiratory tract involvement, pneumonia, convulsive coughing, fever, and cold [15,16,17], accurate diagnosis is necessary. These systemic symptoms are also characteristic of COVID-19, to which gastrointestinal symptoms (e.g., nausea, vomiting, etc.) are sometimes added, while distinctive symptoms specific to SARS-CoV-2 infection include anosmia and ageusia. The symptoms of COVID-19, when present, may be mild or moderate (depending on the presence or absence of pneumonia), becoming severe when the oxygen saturation level falls. In the case of severe disease, the patient presents dyspnea, a respiratory frequency above 30/min, oxygen saturation below 93%, a PaO_2_/FiO_2_ ratio less than 300, and/or lung infiltrates in more than 50% of the lung field within 24–48 h. Finally, respiratory failure, septic shock, and multiorgan dysfunction and/or failure may occur [18]. 

To reduce severe influenza and COVID-19 disease, the only protection is vaccination [19,20]. In Italy, the administration of ‘booster’ doses started on 27 September 2021 as part of the SARS-CoV-2/COVID-19 vaccination campaign [21]. In view of the continuing epidemiological situation regarding the circulation of SARS-CoV-2, the Ministry of Health recommended bringing forward the influenza vaccination campaign to early October 2021 for frail and immunocompromised individuals, subjects aged >65 years, healthcare workers, and children aged between 6 months and 6 years. [22]. The two campaigns therefore overlapped, and, in an effort to increase public awareness and participation in both vaccination campaigns, on 2 October 2021 the Ministry of Health declared the co-administration of the SARS-CoV-2 and influenza vaccines possible [23]. Administering both vaccines during the same visit would have several benefits. For the individual, it would reduce the number of healthcare visits needed and provide timely protection against both diseases; these individual benefits may encourage greater uptake of both vaccines. For health systems, co-administration could facilitate the implementation of both vaccine programs and reduce the overall burden on health services [24]. In addition, co-administration of the two vaccines does not increase immunoreactivity risks [25,26,27]. 

Owing to the characteristics of influenza viruses and the co-circulation of many respiratory viruses, efficient surveillance is necessary. Surveillance aims to clarify the epidemiology of influenza viruses and identify new mutations [28]. The virological surveillance of influenza is an important means of determining the timing and spread of influenza viruses, tracking changes in circulating influenza viruses, and informing seasonal influenza vaccine composition. At the global level, influenza surveillance is conducted by the Global Influenza Surveillance and Response System (GISRS), which is coordinated by the WHO [6,29]. In Italy, surveillance began in 1996 [5] but only became efficient in 2009, during the pandemic caused by the influenza virus A/H1N1 (pdmH1N1/2009). In Italy, the virological and epidemiological surveillance of influenza is carried out by InfluNet [5]. General practitioners or pediatricians carry out European surveillance and collect samples from people affected by influenza-like illnesses (ILIs) or acute respiratory infections (ARIs). 

In 2021, epidemiologic influenza surveillance was carried out from the 42nd week of 2021 to the 17th week of 2022, while virological surveillance was conducted from the 46th week of 2021 to the 17th week of 2022 [30]. Like influenza, COVID-19 presents variable clinical symptoms, ranging from asymptomatic to severe respiratory illness [31]. Therefore, early-stage diagnosis of COVID-19 or influenza is important, since the treatment, pharmaceutical intervention, and prognosis are different [32]. The first aim of this study is to describe the epidemiological and virological status of some respiratory viruses in hospitalized patients affected by SARI during the period 22 November 2021–28 April 2022 in Tuscany, Italy. The second aim is to highlight the importance of respiratory virus surveillance.

## 2. Materials and Methods

### 2.1. Study Design

Oropharyngeal (OP) swabs were collected at the Unit of Emergency Medicine and Internal Medicine II of Santa Maria alle Scotte University Hospital in Siena, Italy from November 2021 to April 2022. Sample collection was conducted in the context of the “DRIVE project”, approved by the Ethics Committee of Area Vasta Sud Est Tuscany, approval Report n. 21,090, on 15 November 2021. Written consent and a questionnaire were obtained from all patients enrolled in the study. DRIVE was set up in July 2017 in order to evaluate Influenza Vaccine Effectiveness (IVE). Since 2020, information on SARS-CoV-2 has also been collected. The objectives and methods of the DRIVE project are described at https://www.drive-eu.org/ (accessed on 9 February 2023) [33]. 

### 2.2. Study Population

All enrolled subjects were affected by SARI. According to the WHO, SARI is an acute respiratory infection that causes symptoms such as coughing and fever and prompts hospitalization within 10 days of presentation. All patients were aged 18+ years, were non-institutionalized, and had not been hospitalized for more than 24 h. 

### 2.3. Specimen Collection

OP swabs were collected from November 2021 to March 2022 from each patient with SARI at Le Scotte Hospital in Siena. The swabs were inserted into a vial containing universal transport medium (D.I.D., Diagnostic International Distribution Spa) and stored at +4 °C in the hospital until they were transported to the Epidemiology Laboratory at Siena University, where they were processed within 24 h. A total of 130 samples were collected, but only 129 samples met the inclusion criteria, which included not being institutionalized and not having been hospitalized for more than 24 h. In addition, the patients studied had to have at least one of the following diseases: cardiovascular disease, lung disease, diabetes, dementia, kidney disease, rheumatologic disease, cancer, obesity, anemia, stroke, and liver disease (shown in Table 1) and at least one respiratory symptom and at least one systemic symptom.

### 2.4. Laboratory Analysis

Viral RNA was extracted from 140 μL samples by means of the QIAamp Viral RNA/DNA Mini Kit (Qiagen, Hilden, Germany). In accordance with the manufacturer’s instructions, RNA was eluted in 60 μL of AVE solution. Viral RNA was rapidly processed by means of real-time reverse transcription-polymerase chain reaction (RT-PCR). We tested the swabs by using RT-PCR FTD Respiratory pathogens 21 (Siemens Healthineers GmbH, Erlangen, Germany), a qualitative test used to identify 21 pathogens, including viruses and some respiratory bacteria: influenza A (IAV); influenza A virus H1N1 (swine strain) (IAV (H1N1) swl); influenza B virus (IBV); human rhinovirus (HRV); human coronavirus (HCoV) 229E, NL63, HKU1, and OC43; human parainfluenza viruses (HPIVs) 1 to 4; human metapneumoviruses (HMPVs) A and B; human bocavirus (HBoV); Mycoplasma pneumoniae (M. pneumoniae); human respiratory syncytial viruses (HRSVs) A and B; human parechovirus (HPeV); enterovirus (EV); and human adenoviruses (HAdVs). RT-qPCR was performed in a final volume of 25 μL in accordance with the manufacturer’s instructions. Cycling conditions were: 50 °C for 15 min, 94 °C for 1 min, and 40 cycles of 8 s at 94 °C and 1 min at 60 °C. The same protocol was applied to SARS-CoV-2 testing (Siemens Healthineers GmbH, Erlangen, Germany). FTD Respiratory pathogens 21 can simultaneously detect the microorganisms listed above owing to the presence of probes and primers that can specifically bind the genome of the microorganism when present in the sample. Very briefly, 5 different master mixes were prepared, each containing enzyme, ddNTP buffer, primer, and probes. Each individual reaction mixture contained multiple primer/probe pairs, and each probe was in turn associated with a fluorescent dye. The dyes were read at the following wavelengths: 520 nm, 550 nm, 610 nm, and 670 nm, each of which is associated with a different microorganism for each individual reaction mixture. In addition, samples that proved positive for influenza A and B were subsequently subtyped for pandemic influenza virus A/H1N1 (Flu A/pH1N1) and seasonal influenza virus A/H3N2 (Flu A/H3N2) and linked to influenza B/Yamagata and B/Victoria, respectively. One-step real-time RT-PCR was performed in a final volume of 25 μL with 0.8 μM forward and reverse primers, 0.2 μM probe, and 5 μL of extracted RNA, in accordance with the manufacturer’s instructions for the use of the One-Step RT-PCR Kit (SuperScript III Platinum One-Step qRT-PCR Kit, Thermo Fisher Scientific, Waltham, MA, USA). Cycling conditions were: 50 °C for 30 min, 95 °C for 2 min, and 45 cycles of 15 s at 95 °C and 30 s at 55 °C. Fluorescence was measured during the 55 °C annealing/extension step. 

## 3. Results

To be enrolled in the study, subjects had to have at least one of the chronic diseases shown in Table 1 and a systemic and respiratory symptom, as shown in Table 2. The median age of the 129 subjects recruited was 76 years (range: 18–101), and more than 50% (68) were female. The most common respiratory symptom was coughing 84%, while only 25% displayed general deterioration. Every subject had at least one chronic disease, the most common (64%) being cardiovascular disease and the least common liver disease (2%), as shown Table 1. Of the subjects enrolled, 40.3% had been vaccinated against influenza; the majority of these (57.7%) had received Fluad Tetra vaccine, while only 9 had received Efluelda. Regarding vaccination against COVID-19, information was collected on the number of doses administered: 105 subjects had received at least 1 dose, while 24 were totally unvaccinated (Table 2).

Of the 129 subjects recruited, 9 (7%, 95% CI 3.2–12) were laboratory-confirmed as being affected by influenza virus A/H3N2. The first influenza-positive subject was found during week 52, corresponding to the first peak of national influenza incidence. Subsequent positives, however, were tested during weeks 9, 11, and 12 at the onset of the second wave of influenza (Figure 1). The median age of the influenza-positive subjects was 55 years (range: 21–83), and 67% (95% CI 29.9–92.5) were male; 89% (95% CI 51.7–99.7) of the positive subjects had not received any influenza vaccine for the 2021/2022 season. Only 1 woman, aged 86, had been vaccinated and tested positive for Flu/A H3N2 121 days after receiving a dose of Fluarix Tetra.

All swabs were simultaneously analyzed for SARS-CoV-2; 55 (43%, 95% CI 34–52) were positive, with most cases occurring in January 2022. A total of 43 subjects (43/80 80%, 95% CI 65–88) had received at least 2 doses of a COVID-19 vaccine. Fifty-four percent of the SARS-CoV-2-positive subjects were female.

We observed 3 co-infections (influenza and SARS-CoV-2), specifically:A 59-year-old male patient, a smoker with chronic respiratory disease. He had received two doses of the Moderna COVID-19 vaccine and one Pfizer booster. He was positive for influenza and SARS-CoV-2 in week 12 of 2022 (21 March 2022). His last vaccination against COVID-19 had been on 8 November 2021.A 23-year-old female patient with diabetes, vaccinated against COVID-19 with 2 doses of the Pfizer vaccine. She was positive by real-time PCR for influenza and SARS-CoV-2 in week 9 of 2022 (3 March 2022). Her last vaccination against COVID-19 had been on 27 November 2021. She had also had COVID-19 a year earlier.A 28-year-old male patient with severe obesity. He had received 2 doses of the Pfizer vaccine against COVID-19. He was positive by real-time PCR for influenza and SARS-CoV-2 in week 9 of 2022 (6 March 2022). His last vaccination against COVID-19 had been on 17 January 2022, and the first on 18 October 2021. He had also had COVID-19 a year earlier.

None of the three patients had been vaccinated against influenza, and all had complained to their physician of similar symptoms, particularly: abrupt onset of symptoms, followed by fever, malaise, muscle pains, and cough.

Furthermore, 2 SARS-CoV-2 and human coronavirus co-infections were detected:A 58-year-old man with diabetes proved positive for HCoV229E. He had been vaccinated against COVID-19 with 2 doses of the Pfizer vaccine and tested positive 6 months after the second dose.A 94-year-old woman with 4 different chronic diseases who had received 3 doses of the Pfizer vaccine tested positive for HCoVOC43 and SARS-CoV-2 3 months after her last vaccination.

Five subjects proved positive for HMPV. Of the 5, an 86-year-old woman was co-infected with SARS-CoV-2, and a 21-year-old man was co-infected with influenza A/H3N2 virus.

HRSV was detected in 2 patients: a 30-year-old female with rheumatic disease who was co-infected with SARS-CoV-2, and a 22-year-old male with diabetes. Only a 75-year-old male was positive for Hpiv-3. No difference in the incidence of cases of co-infection was detected between males and females.

## 4. Discussion

In March 2020, the WHO declared a global state of emergency due to the presence of a new coronavirus, now known as SARS-CoV-2. As the virus began to spread rapidly, non-pharmacological interventions (NPIs) were applied worldwide and helped bring the spread of the virus under control [34]. The measures to control the transmission of infections were: the use of masks, social distancing, working from home, limiting gatherings, and increasing hygiene measures. These interventions, which were aimed at limiting the transmission of SARS-Cov-2, also had an important impact on other airborne viruses, such as influenza viruses, measles, rubella [35], respiratory syncytial virus, common human coronaviruses (HCoVs), parainfluenza viruses, and metapneumoviruses [36,37]. However, since most laboratories were busy battling the scourge of SARS-CoV-2, it is likely that the decrease in circulation was also compounded by the underdiagnosis of other airborne viruses [36]. In Italy, the 2020/2021 influenza season was characterized by cases of ILI (influenza-like illness), but these were always below the threshold, and no positive cases of influenza were reported. 

The flu season displayed a similar pattern in the winter of 2009/10; after an initial pandemic wave during the 45th week, the incidence rate fell to 12.5 per 1000 population, and few cases of influenza were reported. This trend is identical to that seen in Italy during the 2020/21 flu season, as a result of the pandemic wave due to SARS-CoV-2 (Figure 1).

The 2021/2022 season also saw exceptionally low circulation of the influenza virus. The first cases were observed during the 52nd week of 2021; subsequently, from the 8th week of 2022 onwards, circulation increased, reaching its peak in the 12th week of 2022, when the COVID-19 restrictions were eased [38]. This peak, however, never reached the levels frequently observed before the SARS-CoV-2 pandemic.

In Italy, influenza vaccination coverage among people older than 65 years decreased to 58.1% in 2021/22 from 65.3% in the previous season. In our study population, 40.3% of subjects were vaccinated against influenza, compared with 20.5% of the general Italian population and 21.8% in the Tuscany region [39]. In March 2022, administration of the fourth dose of COVID-19 vaccine began in Italy, but high coverage was never reached. Similarly, in the same period, the administration of booster doses (third dose) declined; this was probably because the so-called “green pass”, which was required to access many normal activities, was starting to be phased out [40]. During the 2021/2022 season, the A/H3N2 influenza virus predominated in Italy. Accordingly, all nine swabs that tested positive for influenza in our study were positive for influenza A/H3N2 virus. Moreover, apart from the first case, which occurred in the 52nd week of 2021, 8 were isolated in mid-March 2022, reflecting the epidemiological trend of influenza viruses in Italy and the public’s compliance with COVID-19 vaccination.

The three co-infections of SARS-CoV-2 and influenza A/H3N2 also occurred in March 2022. These three subjects had not been vaccinated against influenza, and it is probable that the increased expression of Angiotensin-converting enzyme 2 (ACE-2) following influenza virus infection also promoted the SARS-CoV-2 infection [41]. The ACE-2 receptor is most highly expressed in the epithelial cells of the upper respiratory tract, whereas it is present at lower levels distally. Following IAV infection, it appears that alveolar type II (ATII) cells in the lung alveoli increase the expression of ACE-2 mRNA. As the spike protein of SARS-CoV-2 binds the ACE-2 receptor and induces increased transcription, when SARS-CoV-2 interacts with IAV, its access is also facilitated in the distal regions of the lung, thereby exacerbating COVID-19 symptoms [42].

In addition, only 29% of our influenza vaccinees tested positive for COVID-19; this was probably due to trained immunity, i.e., the innate immune system is ready to face a challenge by the same or another microorganism [43]. It is also interesting that the influenza-positive SARI patients were younger than those who were positive for COVID-19. Finally, regardless of virus mismatch versus vaccine composition, only 1 influenza-positive subject had been vaccinated against influenza; the other 8 had not. As mentioned above, co-infection of hMPV and SARS-CoV-2 exacerbates the clinical manifestation of the latter. Indeed, the woman who tested positive for both viruses presented at the hospital with several systemic manifestations and all respiratory manifestations except for sore throat. The other patients also had worse symptoms than those who had been infected with only one virus.

Despite the obvious limitations imposed by the low number of subjects enrolled, our study emphasizes the well-known importance of infectious disease prevention. Primary prevention is even more imperative, given the known likelihood of co-infection by microorganisms against which vaccines have not yet been formulated. Vaccination against one microorganism goes a long way toward alleviating symptoms and reducing the risk of complications, especially in vulnerable population groups, such as those in our study. Indeed, all our subjects had at least one chronic disease, which exacerbated symptoms in subjects infected with more than one virus [44]. Our study has some limitations. First, as the sample size was limited by the overall availability of swabs collected, it may not have been fully representative of the population. Although the study was conducted in a 2nd-level hospital with about 700 beds, and the specific catchment area of the hospital has around 120,000 inhabitants as a reference for basic activities, only 2 departments participated in the study. In addition, the low number of influenza cases in the 2021/2022 season also had an impact.

## 5. Conclusions

Our results demonstrate the importance of carrying out epidemiological and virological surveillance of influenza and other respiratory microorganisms, as this practice provides beneficial information that physicians can utilize in order to ensure appropriate treatment and diagnosis. Vaccination remains the most effective method of preventing severe forms of respiratory diseases in individuals at high risk of hospitalization and complications. It is very important to monitor the progress of viruses other than SARS-CoV-2, so as not to find ourselves unprepared to face other possible pandemic viruses, and sensitive differential laboratory diagnosis helps to avoid subjecting individuals to unsuitable treatment. In conclusion, constant monitoring of virological sequences circulating in the territory still remains the best means of ensuring the formation of up-to-date vaccines in the shortest possible time.

## Figures and Tables

**Figure 1 vaccines-11-00776-f001:**
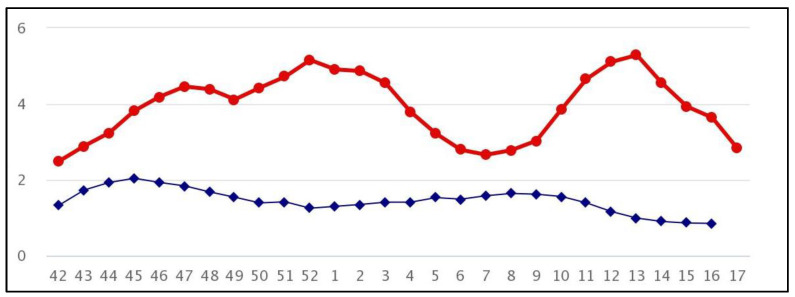
Influenza incidence in Italy, seasons 2020/2021, 2021/2022: The figure shows the incidence, per 1000 people, of influenza cases in the Italian population in the 2020/21 (in blue) and 2021/22 (in red) seasons. The abscissae show the weeks of epidemiological surveillance, and the ordinates show the incidence rates.

**Table 1 vaccines-11-00776-t001:** Chronic diseases in the study populations: number of subjects (N), prevalence (%), and confidence interval at 95% (CI 95%).

Chronic Disease	N° (%)	CI 95%
Cardiovascular disease	82 (64)	54.7–71.9
Lung disease	47 (36)	28.1–45.4
Diabetes	39 (30)	22.5–39
Dementia	17 (13)	7.9–2
Kidney disease	12 (9)	4.9–15.7
Rheumatologic disease	9 (7)	3.2–12.9
Cancer	7 (5)	2.2–10.9
Obesity	5 (4)	1.2–8.8
Anemia	3 (2)	0.5–6.7
Stroke	3 (2)	0.5–6.7
Liver disease	2 (2)	0.2–5

**Table 2 vaccines-11-00776-t002:** Characteristics of the population enrolled. For each characteristic, the total number (N), prevalence (%), and confidence interval at 95% (CI%) are reported.

Median Age		76 years	
Range		18–101	
Gender		N (%)	CI 95%
	Males	61 (47)	38.44–56.26
	Females	68 (53)	43.7–61.6
Respiratory symptom		N (%)	CI 95%
	Cough	109 (84)	77–90
	Breathing difficulties	90 (70)	61–77.5
	Sore throat	33 (26)	18.3–34
Systemic symptom		N (%)	CI 95%
	Fever	122 (95)	89.14–97.8
	Malaise	122 (95)	89.14–97.8
	Abrupt onset	118 (91)	85.25–95.7
	Headaches	94 (73)	64.34–80
	Myalgia	90 (70)	61–77.5
	Deterioration	32 (25)	17.63–33.18
COVID-19 vaccination		N (%)	CI 95%
	Total	105 (81,4)	73.6–87.7
	1 dose	9 (8.5)	4–15
	2 doses	28 (26.7)	18.51–36
	3 doses	68 (64.8)	54.9–73.84
	Unvaccinated	24 (18.6)	12.3–26
Influenza vaccination		N (%)	IC 95%
	Total	52 (40.3)	31.8–49.3
	Fluad Tetra	30 (57.7)	43.2–71.2
	Fluarix Tetra	13 (25)	14–39
	Efluelda	9 (17.3)	0.8–3

## Data Availability

Further details and results from the DRIVE studies are available at the DRIVE website via the following link: https://www.drive-eu.org/ (accessed on 6 March 2023).

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
