# Peer review of "Surveillance of Influenza and Other Airborne Transmission Viruses during the 2021/2022 Season in Hospitalized Subjects in Tuscany, Italy"

_vaccines, 2023, doi:10.3390/vaccines11040776_

Round 1

Reviewer 1 Report

Dear Editor-in-Chief

The manuscript in title of” Surveillance Of Influenza And Other Airborne Transmission 2 Viruses During The 2021/2022 Season In Hospitalized Subjects 3 In Tuscany, Italy” shows the importance of respiratory virus infection at the peak of the viruses circulation.

There are some sentences which need to be improved such as:

-The keywords must be unified in capitalization.

-Line 142 “Each subject enrolled” the word order must be cheched.

-Line 153 states that “A total of 130 samples ….. had the right inclusion criteria.” Please explain the inclusion criteria in the methods.

-Line 209:“during week 52, during the first peak” needs to be refined.

The descriptive data in the methods section is too much in written form.  A summary form of the presented cases in a table could help the readers to compare the results.

Reviewer 2 Report

The article lacks original ideas and may not be suitable for publication.

Reviewer 3 Report

v                 Suggestions to Author/s

1. Dear Dr. Giovanna Milano, as a selected reviewer, I made the prompt check of your excellent article:” Surveillance of Influenza and Other Airborne Transmission Viruses during the 2021/2022 Season in Hospitalized Subjects in Tuscany, Italy” and found it:(X) Excellent, accept the submission (5).

2. During the prompt check, at first the pdf was converted to doc, so that the corrections became possible. Some small mistakes were found, and they are highlited into red colour. You are kindly asked to correct them so that you changed them into black colour.

3. Finnaly, you are kindly asked to arrange the final English language before the final editing of the article.

Round 2

Reviewer 2 Report

It is evident that the author has made sincere efforts to revise the paper. However, I still feel that the paper lacks novelty and significance. Nevertheless, considering that the author has diligently incorporated the feedback of other reviewers, I have no further comments.